# A Rapid Method for Detection of Antigen-Specific B Cells

**DOI:** 10.3390/cells12050774

**Published:** 2023-02-28

**Authors:** Mariia Vakhitova, Mikhail Myshkin, Dmitriy Staroverov, Irina Shagina, Mark Izraelson, Nadezhda Tverdova, Olga Britanova, Ekaterina Merzlyak

**Affiliations:** 1Shemyakin–Ovchinnikov Institute of Bioorganic Chemistry, Russian Academy of Sciences, 117997 Moscow, Russia; 2Institute of Translation Medicine, Pirogov Russian National Research Medical University, 117997 Moscow, Russia

**Keywords:** SARS-CoV-2, BCR-repertoire, B cells specific response to COVID infection

## Abstract

The global SARS-CoV-2 pandemic has united the efforts of many scientists all over the world to develop wet-lab techniques and computational approaches aimed at the identification of antigen-specific T and B cells. The latter provide specific humoral immunity that is essential for the survival of COVID-19 patients, and vaccine development has essentially been based on these cells. Here, we implemented an approach that integrates the sorting of antigen-specific B cells and B-cell receptor mRNA sequencing (BCR-seq), followed by computational analysis. This rapid and cost-efficient method allowed us to identify antigen-specific B cells in the peripheral blood of patients with severe COVID-19 disease. Subsequently, specific BCRs were extracted, cloned, and produced as full antibodies. We confirmed their reactivity toward the spike RBD domain. Such an approach can be effective for the monitoring and identification of B cells participating in an individual immune response.

## 1. Introduction

Research on the T and B cells that mediate the immune response to SARS-CoV-2 infection is associated with technical challenges. The availability of an enormous number of biological samples from patients with various levels of disease severity and the accumulation of sequencing data have prompted the development of strategies for the detection of antigen-specific B cells.

The primary target for vaccine and therapeutic-antibody development is the SARS-CoV-2 spike (S) protein, which facilitates host cell attachment and penetration. Spike consists of three domains, where RBD serves as the dominant target for neutralizing antibodies that prevent the binding of RBD to ACE2. A good prognosis in terms of COVID-19 patient survival correlates with the existence of anti-RBD IgG antibodies. Mutations in the SARS-CoV-2 genome as a whole, and in the RBD domain in particular, accumulate rapidly. For example, common mutations in the RBD residues K417 and E484 significantly reduce the neutralization activity of serum from convalescent and vaccinated subjects and abrogate the neutralization activity of many monoclonal antibodies [1]. This tendency requires the design of novel approaches to finding new antibodies against SARS-CoV-2 antigens within a short time frame. Methods that provide access to potentially neutralizing or antigen-specific immunoglobulin (Ig) sequences have been sufficiently improved for the recent COVID-19 pandemic, but high cost and difficulties with the implementation of most protocols can hamper the discovery of SARS-CoV-2–specific B-cell receptors (BCRs).

The widely used analysis of the bulk Ig repertoire in peripheral blood mononuclear cells (PBMCs) [2,3,4], or in sorted B-cell subsets, allows for the estimation of isotype frequency, somatic hypermutations, V gene usage, and clonality. For the last decade, various tools have been developed for Ig repertoire profiling [5]. The relevant samples may be collected from patients after SARS-CoV-2 infection or after vaccination and compared to the samples from healthy donors. A downstream bioinformatics analysis may reveal clusters of sequences indicative of a SARS-CoV-2–related immune response. Additionally, the CoV-AbDab database can be used to confirm the specificity of the sequences potentially associated with a response to SARS-CoV-2 antigens. A major drawback of bulk B-cell receptor mRNA sequencing (BCR-seq) is that neither the antigenic specificity nor the neutralizing capacity of the antibodies encoded by the identified BCR sequences can be determined. Moreover, data regarding individual heavy- and light-chain pairs is missing due to the nature of this approach. Heavy and light chains of interest can potentially be paired based on their frequencies, but due to the low diversity of light chains, this technique rarely works.

Another approach involves the VDJ 10X Genomics analysis of a B-cell subset isolated from PBMCs [6]. The main advantage of this methodology is that it yields a paired repertoire of heavy and light BCR chains. Nonetheless, the low number of analyzed BCRs can barely cover the Ig repertoire diversity and provides information only about the most abundant clonotypes. An attempt to increase the cell number in this analysis dramatically raises the cost of the experiment.

A combination of the latter two approaches (deep BCR repertoire profiling and 10X Genomics single-cell sequencing) offers comprehensive data on individual heavy- and light-chain pairs and heavy-chain clusters. Single-cell sequencing may be employed in two modes: single-cell transcriptome sequencing (scRNA-seq, sequencing of all mRNAs) and single-cell BCR sequencing (scBCR-seq, sequencing of only *BCR* mRNAs). For example, in ref. [7], a strong S protein–binding activity was demonstrated for 14 extracted antibodies out of 347 selected heavy- and light-chain pairs.

In a collaborative study [8] conducted by several medical centers across the United Kingdom, data on transcriptomic and surface markers (CITE-seq) were used to devise a multi-omics single-cell approach. Nevertheless, in the case of B cells, and especially plasma cells, such a methodology is not efficient owing to the low abundance of these cell types in peripheral blood.

The most promising techniques comprise enrichment of cell samples with individual antigen-specific B cells by cell sorting using SARS-CoV-2 proteins as labeled targets [9]. In a variation of this method [10], B cells are incubated for several days with CD40/IL-21/BAFF, followed by VDJ 10X Genomics single-cell sequencing. This method, however, has a very low yield, and a substantial amount of data is lost in the process. It involves rather challenging cultivation of B cells and considerable costs of 10X Genomics for high-throughput screening.

One of the more complex techniques combines BCR-seq data on peripheral-B-cell antibody variable-region repertoires with plasma IgG proteomic sequencing by tandem mass-spectroscopy (Ig-seq) [11]. Because plasma IgG can be affinity purified prior to this procedure, this experimental design can generate detailed information about the plasma memory BCR repertoire and about RBD-specific BCRs.

An attempt to overcome the low abundance of RBD-specific B cells in peripheral blood was made in a model experiment [12] where mice were vaccinated with lipid nanoparticle-encapsulated *RBD* or *S* mRNA. Eight weeks after the vaccination, SARS-CoV-2–specific germinal-center memory B cells (GL-7^+^, B220^high^, CD38^low^, IgD^−^, CD93^−^, CD138^−^, and SARS-CoV-2 RBD-specific or S-specific) from lymph nodes were sorted by flow cytometry and subjected to 10X Genomics for high-throughput VDJ sequencing. The results showed that in the RBD group, all 100 monoclonal antibodies were expressed; 78 (78%) of them bound RBD, and 63 (63%) showed neutralizing activities. This method has been applied to humans as well, by means of draining axillary lymph nodes, PBMCs, and bone marrow cells [13]. In that study, the evolution of 1540 spike-specific B-cell clones was tracked via a combined approach: a BCR-seq analysis of responding blood plasmablasts and memory B cells, lymph node germinal-center B cells, and plasma cells and bone marrow plasma cells from eight individuals; expression of the corresponding monoclonal antibodies common between plasma cells and bone marrow plasma cells revealed high antigen specificity and neutralizing activity.

IgG and IgM antibodies against the spike protein are detectable in a substantial proportion of COVID-19 patients in the post-acute phase. On the other hand, a high titer of neutralizing antibodies is frequently associated with severe cases of COVID-19 [14]; this state of affairs increases the probability of capturing memory B cells in such samples. In this work, we present a method that integrates the sorting of antigen-specific B cells from PBMCs of patients with severe acute COVID-19, BCR-seq, and computational analysis, enabling us to find BCR-specific sequences. This fast and cheap technique may be useful in various fields when the antigen is known and a specific BCR sequence has to be identified for subsequent analysis.

## 2. Materials and Methods

### 2.1. Patients and Samples

COVID-19 patients were hospitalized at the intensive care unit of the Federal Center of Brain Research and Neurotechnologies, the Federal Medical Biological Agency, and the National Medical Research Center for Hematology, Moscow, Russian Federation.

The patients met the following criteria: (1) All COVID-19 subjects were confirmed positive according to quantitative reverse transcription PCR (RT-qPCR); (2) severe COVID-19 manifestation was determined according to NIH Coronavirus Disease 2019 Treatment Guidelines, 10–14 days after symptoms appeared in the acute phase. Peripheral blood samples from four patients were obtained: F 38 (1), M 31 (2), F 22 (3), F54 (4).

The study protocol was approved by the PRNRMU Ethic Committee #221. The study was conducted according to the International Conference on Harmonization Guidelines for Good Clinical Practice and the Declaration of Helsinki 1964, along with its later amendments.

### 2.2. B Cell Isolation and RBD-Specific B Cell Enrichment

PBMCs were isolated from 20 mL of peripheral blood samples. Blood samples were diluted 4 times with Hanks, layered on Ficoll-Paque (Paneko, Moscow, Russia), and spun at 500× *g* for 30 min at room temperature. The leukocytes were then washed in Hanks, and centrifuged at 300× *g* for 15 min at room temperature. The cells were stained with anti-CD-19-PE (Biolegend, San Diego, CA, USA), which is a specific marker for all B lineage cells, then incubated with anti-PE beads. Magnetic separation was performed using ML columns (Miltenyi, Auburn, CA, USA), according to the manufacturing protocol. The purity of CD19 positive magnetic separation was estimated as 80–90% (Figure A1). After positive selection, the B cells were additionally stained with CD27-BV421 (BioLegend, San Diego, CA, USA) and RBD-Alexa647 (in house), which is widely used as a marker for memory B cells. RBD protein was kindly provided by Dr. G. Efimov (National Research Center for Hematology, Moscow, Russia), and labeled in house with Alexa 647 (Alexa Fluor™ 647 Microscale Protein Labeling Kit, Invitrogen, Thermo Fisher Scientific, Inc., Waltham, MA, USA), according to the manufacturing protocol.

The sorting was maintained with the Sony SH800S Cell Sorter. Three populations were sorted, CD19^+^CD27^+^RBD^+^, CD19^+^CD27^−^RBD^+^, and CD19^+^CD27^+^RBD^−^ (as a negative population), in the RLT buffer (Qiagen GmbH, Hilden, Germany) to stabilize the RNA, and then stored at −20 °C.

Total RNA was isolated with the RNeasy MinElute kit (Qiagen GmbH, Hilden, Germany). The cDNA library was constructed with the SMART-Seq^®^ v4 Ultra^®^ Low Input RNA Kit for Sequencing (Takara Bio, Inc., Otsu, Japan). RNASeq libraries were prepared for scarcelow-abandoned samples (less than 500 cells). For these samples, heavy chains were amplified with Human Ig RNA Multiplex (MiLaboratory Inc., San Francisco, CA, USA) using RNASeq library as a template. For rich samples (more than 500 cells), heavy chains were amplified with Human Ig RNA (MiLaboratory Inc., San Francisco, CA, USA) using cDNA as a template. All samples were barcoded with Nextera XT (Illumina Inc., San Diego, CA, USA) indexes and sequenced with the Illumina MiSeq system using 300 + 300 cycles kit.

### 2.3. Computational Analysis

Both the Human IG RNA Multiplex Kit (MiLaboratory Inc., San Francisco, CA, USA) and the Human BCR kit (MiLaboratory Inc., San Francisco, CA, USA) use unique molecular identifiers (UMIs) for error-correction to determine precise coverage and clone count. The MiNNN tool was used to extract molecular barcode sequences, correct sequencing and/or amplification errors, and assemble reads with the same UMI into molecular identifier groups (MIGs). MIGs containing only one read were excluded from further analysis. Assembled sequences were analyzed with the MiXCR tool (MiLaboratory Inc., San Francisco, CA, USA) [15] to extract BCR clones. Samples without molecular barcodes (UMI) were assembled into clones directly from raw reads using the MiXCR tool. The analysis of diversity and the CDR3 properties of BCR clones was performed using VDJtools software (MiLaboratory Inc., San Francisco, CA, USA) [16]. For diversity and Shannon–Wiener index calculations, the repertoires were downsampled to 1100 reads/MIGs. The clusterization of IGH, IGK, and IGL chains was analyzed using in-house python script and visualized using Cytoscape software (https://cytoscape.org/).

### 2.4. Data Availability

B cell repertoire sequencing data were deposited in the SRA database under project number PRJNA885120.

### 2.5. Cloning

The extracted sequences that form clusters were manufactured by the Cloning Facility (Cloning Facility, Moscow, Russia). Heavy and light chains were cloned in the pFUSE-CHIg-hG1 and pFUSE2-CLIg-hK vectors (InvivoGen, San Francisco, CA, USA), respectively. Cloning was maintained using the cyclone method [17]. Cloned insertions were verified by Sanger sequencing (Evrogen, Moscow, Russia). Then purified plasmids were transfected into CHO cells and expressed in serum-free medium ExpiCHO™ (Thermo Fisher Scientific, Inc.). The ExpiFectamine™ CHO Transfection Kit (Thermo Fisher Scientific, Inc.) was used for transfection. Immunoglobulins were purified from the cultivation medium on HiTrapTM Protein A HP columns (GE Healthcare AS, Oslo, Norway), according to the standard protocol. Purified immunoglobulins were verified with 12% denaturing SDS-PAGE in native and reduced conditions. ELISA was subsequently used in two dilutions of antibodies (10 mkg and 1 mkg per cell) using RBD immobilized plates (SARS-Cov-2-IgG-ELISA National Medical Research Center for Hematology, Moscow, RF, and RBD-immobilized plates were made elsewhere, accordingly standard protocols). Immunoglobulin free BSA (GeneTex Inc., Alton Pkwy Irvine, CA, USA) served as a negative control of immobilization, the serum of vaccinated donors was used as a positive control, which was additionally confirmed by a medical facility, and monoclonal antibody with another specificity was used as a control for RBD binding. ELISA binding to SARS-CoV-2 RBD was measured by absorbance at 450 nm. Antibody concentrations starting at 0.4 mg/mL were used and diluted 10-fold. Such measurements were maintained in triplicate in different levels of immunoglobulin purification.

## 3. Results

### 3.1. Sorting of RBD-Specific B Cells

With the help of a fluorescently labeled RBD protein for the sorting of CD19^+^ cells, we obtained the following cell fractions: RBD^+^ plasmablast and memory B cells (CD19^+^CD27^+^ RBD-specific), RBD^+^ naïve B cells (CD19^+^CD27^−^ RBD-specific), and other plasmablast/memory B cells (CD19^+^CD27^+^ RBD-nonreactive; Table A1, Figure A1). In all four samples, RBD^+^ cells represented 0.19–2.40% of all memory B-cell subsets, while the proportion of memory B cells was 49–56%. Thus, the percentage of RBD^+^ B cells among CD19^+^ population varied in the 0.45–1.60% range among the examined samples, in agreement with previously published findings [18,19].

CD27^+^ B cells are considered a classic memory B-cell subset. On the other hand, a skew of B-cell populations in COVID-19 patients has been documented [20], i.e., a 2–3-fold higher proportion of atypical memory B cells having the CD21^low^CD27^−^CD10^−^ phenotype and a much lower proportion of the classic memory B-cell population as compared to healthy donors and convalescent patients. We cannot rule out that the RBD^+^ B cells in the CD27^−^ population were in fact, naïve and not atypical memory B cells; this could be verified by using additional markers such as CD21 and CD10. In other articles, different populations of double-negative (CD27^−^IgD^−^) B cells have been distinguished, and it is reported that they result from extrafollicular maturation [21,22]. B cells with hallmarks of extrafollicular maturation often appear in critically ill patients with COVID-19. Such B-cell responses strongly correlate with high concentrations of SARS-CoV-2–neutralizing antibodies and poor clinical outcomes. Additionally, RBD-specific B cells have been found among naïve B-cell populations in donors who tested negative for COVID-19 [23]. Accordingly, we pooled the repertoire data from RBD^+^ memory and RBD^+^ naïve B-cell populations for a subsequent comparison with RBD^−^ B cells.

### 3.2. Computational Analysis of BCR Repertoires for Extraction of RBD-Specific BCR Clusters

The repertoires of heavy and light BCR chains were extracted by means of the MiXCR tool [15]. The numbers of clonotypes are presented in Table A2.

We estimated the diversity and evenness of BCR heavy chains in the repertoires from three sorted B-cell populations, as well as the mean nucleotide sequence length of the *CDR3* regions (Figure 1). We found no significant differences in diversity, evenness (Shannon–Wiener index), *CDR3* length, or hydropathy (kf4) between the RBD^+^ and RBD^−^ B-cell populations. Nevertheless, the RBD^+^ B-cell populations tended to be less diverse and less even, according to the Shannon–Wiener index.

In all sorted groups of B cells, we observed the prevalence of the IgM isotype, but this prevalence was significantly higher (*p* = 0.011, Wilcoxon test) in the RBD^+^ B-cell population than in the RBD^−^ group, accounting for 95% of the repertoire (Figure 2). This finding is consistent with numerous studies in which a response via a secreted SARS-CoV-2–specific IgM antibody occurred and peaked earlier than that of IgG [24], and anti-RBD antibodies initially proved to be IgM [25].

Moreover, it has been demonstrated in phage display experiments that in the first days of the infection, IgM antibodies from infected donors recognize diverse epitopes of SARS-CoV-2 antigens, including RBD, while IgG and IgA recognize fewer or no epitopes [26].

As described in numerous COVID-19 studies [3,4,6,9,27], VDJ gene segments from antigen-specific sequences are characterized by preferential usage of the Ig heavy chain variable region 3 (*IGHV3*) subfamily genes, mostly rearranged with the Ig heavy chain joining region 4 (*IGHJ4*) or *IGHJ6* gene segments [28]. The repertoire of the RBD^+^ B-cell population in our work was also found to be enriched with *IGHV3*, *IGHJ5*, and *IGHJ6* (Figure 2B,C). We detected no significant increase in *IGHV3-53* usage, but noticed enrichment with some other genes: *IGHV2-5*, *IGHV3-23*, and *IGHV4-59* (Figure 2B). The heterogeneity of V usage seen in our patients’ samples suggested that the response to SARS-CoV-2 was not driven exclusively by the specific *IGHV* genes, in agreement with [27].

Next, we performed a detailed analysis of the *CDR3* region. We observed a trend of greater *CDR3* region length in clustered *IGHV* RBD^+^ sequences in comparison to RBD-nonreactive samples (Figure 1C). This result is supported by several studies [6,27], in which the effect is even stronger for severe cases of COVID-19, but in other research articles, *IGH CDR3* length does not show a significant change [29].

### 3.3. Clusterization of Heavy Chains

In all the sorted B-cell subsets, we found clusters of BCR heavy chains that had identical *IGHV* and at most, one amino acid mismatch in the *CDR3* region. In our analysis, only RBD^+^ B cells formed clusters with more than three nodes (Figure 3A) and contained convergent virus-specific antibody sequences shared among the COVID-19 patients. Similar results have been obtained in multiple other COVID-19 studies [30]. Most of our clusters predominantly contained *IGHV* sequences from one patient with a partial overlap with the samples from other patients. Such a large overlap suggests that the response to COVID-19 is similar among patients in terms of the V gene segment and *CDR3* region of the heavy chains. Unlike the RBD^+^ samples, most of our RBD^−^
*IGHV* sequence chains turned out to be singletons and at best, the clustered sequences had only three nodes. Such a cluster distribution implied that the sorted B-cell populations were indeed enriched with antigen-specific B cells.

Apparently, we were unable to eliminate contamination of the enriched RBD^+^ clonotypes with the Alexa Fluor 647–specific type. Double tagging of the antigen with two different dyes will undoubtedly decrease the impact of such nonspecific signals. Due to the very small number of registered events in our experiment, such an approach could diminish them even further. At the next step, when choosing the candidate RBD^+^ sequences, we utilized the following strategy to select the candidate clonotypes that potentially react with RBD:

-filter out singleton clonotypes, -filter out the most abundant clusters of *IGH CDR3*s, assuming their nonspecificity to RBD.

Finally, we selected four clusters with high prevalence node counts in one patient and several overlaps with the other donors. In every cluster, the sequence with the highest clonotype count was chosen as the most promising for subsequent cloning. Examples of such clustered clonotypes are presented in Figure 2B. Additionally, we detected a large number of CDR3s containing poly-tyrosine stretches. We did not find any literature regarding the biochemical properties of COVID-19 + BCR amino acid sequences. Nonetheless, it has been demonstrated that polyreactive and autoreactive IgM natural antibodies encoded by the V regions having high contents of tyrosine and serine residues possess higher binding flexibility [31]. Furthermore, an analysis of a germ line contribution to CDR3 diversity indicates that loops from naïve B cells are dominated by Tyr and small residues (Gly, Ser, Ala, and Thr) [32]. Although the high abundance of Tyr residues in CDR3 is typically a feature of a naïve repertoire, several neutralizing RBD^+^ antibodies have a high Tyr content, i.e., LY-CoV555 [33]. Therefore, we decided to include such high-Tyr clones in our assay.

To evaluate the number of hypermutations in the *IGH* clonotypes, we compared the nucleotide *IGHV* sequences (excluding *CDR3* regions) with the corresponding germline samples and determined the mutation rate per base pair. Clonotypes from larger clusters (four or more clonotypes) had significantly fewer hypermutations per base pair (*p* < 2.2 × 10^−16^) than did clonotypes from smaller clusters (1–3 clonotypes; Figure 4).

This observation has several explanations: 

-most of the cells in the RBD^−^ population are mature memory cells, which accumulate a large number of hypermutations;-the RBD^+^ B-cell subset can be divided into two groups of clones: those from smaller clusters with a high hypermutation rate, and clones from large clusters with near-germline sequences. These findings point to two types of RBD response: a memory response based on the cells left after previous coronavirus infections, and a naïve primary response to SARS-CoV-2 proteins.

### 3.4. Light-Chain Analysis

Light chains were also processed by the multiplex protocol; kappa and lambda isotypes were extracted (Table A2). Both chains were present in the extracted repertoire of the light chains.

Elevated frequencies of families *IGKV1*, *IGKV2*, and *IGKV3* were observed in the kappa chain, with the main V genes being *IGKV3-20*, *IGKV3-11*, *IGKV2-30*, and *IGKV1-39* (Figure 5A). *IGLV3* dominated in the lambda chains, with high frequencies of the use of *IGLV3-19* and *IGLV3-21* segments (Figure 5B). Low clonotype diversity and a complicated network of light-chain overlaps in the cluster analysis prevented us from extracting “proper” sequences associated, for instance, with individual donors. For pairing with RBD^+^ IGH, we selected individual light-chain clonotypes derived from two large clusters of RBD^+^ B cell fractions built from clonotypes of two unrelated donors (Figure A2), which were expanded in the repertoire. These IGK chains were homologous to the public IGKV3-20 and IGK1-39 identified as a part of the neutralizing antibody in other patients [27]. Notably, Cov-Ig1,2 (Table A3) and CDR3 (QQYGSSLWTF) contained amino acids Y(92) and W(98) that were essential for Ag recognition [34]. The HC-LC pair in Cov-Ig1 (IgHV3-23 IgKV3-20) also occasionally coincidence with one of the five most frequently paired heavy and light chain clonotypes [9]. The third chain was selected from a small cluster where the light chain was expanded in the IGK repertoire of the third donor.

### 3.5. Ig Expression

The heavy chains from clustered clonotypes and overpresented light chains were chosen for cloning (Table A3) into pFUSE vectors under control of the Elongation Factor-1α (EF-1α) core promoter. The recombinant plasmids were transfected in CHO cells, which were grown in serum-free media, and the secreted antibodies were purified from the culture medium on Protein A Sepharose beads. SARS-CoV-2 RBD specificity was tested on commercial or in-house immunological plates. A positive signal was obtained with all four tested antibodies (Table A4). Remarkably, the Cov-Ig-3 antibody that showed the highest absorbance value in ELISA compared to other antibodies originated from the largest cluster (Figure 3B).

## 4. Discussion

Here, we propose an approach that consists of several key steps: RBD-specific sorting of B-cells, BCR repertoire sequencing, and subsequent bioinformatics analysis. Based on downstream repertoire analysis, we can conclude that RBD-specific BCRs mainly form the largest clusters and can be shared between donors. These features indicate specific clonotypes for selection and subsequent expression in mammalian cells. 

In repertoire analysis, we observed the heterogeneity of V-gene usage that might be related to the individual response to SARS-CoV-2. As demonstrated previously, V-gene usage variation among individuals was far larger than differences among disease severity cohorts [25]. Additionally, viral subvariants of SARS-CoV-2 elicit immune response mediated by antibodies utilizing different IGHV gene segments [18].

Despite the small size of the cohort and the heterogeneity of V usage, the proposed clustering analysis of Ig clonotypes from sorted B cells proved to be helpful for finding SARS-CoV-2-specific Ig sequences.

This approach can be applied to other tasks that involve a search for antigen-specific B cells. One of the main problems with finding antigen-specific B cells in peripheral blood is their very low number. We managed to find RBD^+^ B cells among PBMCs due to the acute stage of infection, when the number of specific cells is the highest. Thus, a limitation of this method for finding antigen-specific B cells is that the donors should be at the acute infection stage. Besides the low abundance of specific memory B cells and plasmatic cells in peripheral blood, high similarity and cross-individual sharing of light immunoglobulin chains are also obstacles to the functional pairing of immunoglobulin chains. Nevertheless, the proposed technique may be more applicable to diseases with a sufficiently high local antigen load, i.e., a tumor microenvironment (which is often infiltrated by B cells) or sites of chronic inflammation. 

## Figures and Tables

**Figure 1 cells-12-00774-f001:**
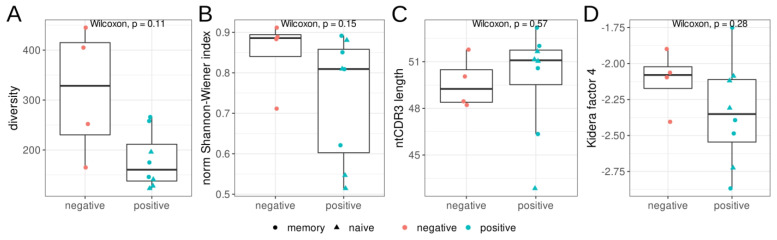
Features of the heavy–chain (*IGH*) repertoires of RBD^+^ (red) and RBD^−^ (teal) B cells isolated by cell sorting from donors with severe COVID-19 in the acute phase. Triangles represent CD27^−^ naïve memory B cells, and circles show CD27^+^ memory B cells. (**A**): Observed diversity of repertoires, (**B**): the normalized Shannon–Wiener index, (**C**): the mean nucleotide length of the *CDR3* region. For diversity and Shannon–Wiener index calculations, the repertoires were downsampled to 1100 reads/MIGs, (**D**): mean Kidera factor 4 (kf4) for full CDR3 amino acid sequences. Positive values of kf4 are characteristic of charged and polar residues, negative values are hydrophobic.

**Figure 2 cells-12-00774-f002:**
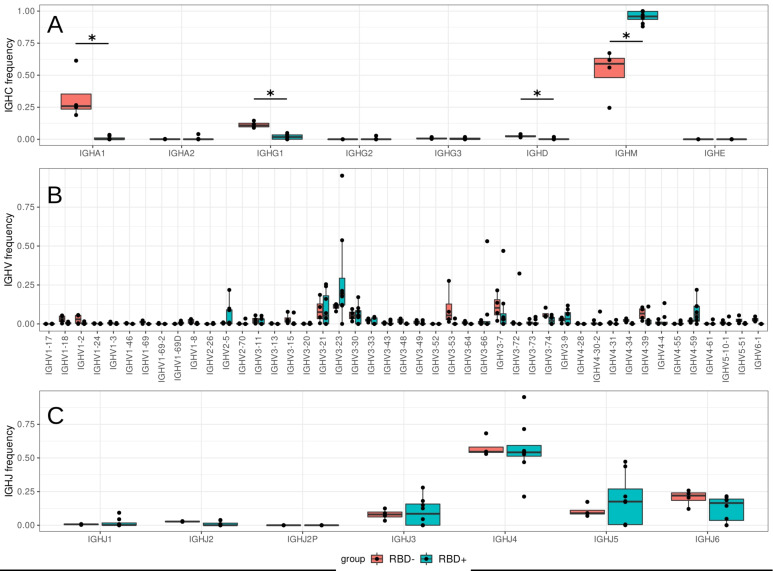
The proportion of usage of Ig isotypes (**A**) and V (**B**) and J genes (**C**) in the sorted B-cell subtypes from four patients (*n* = 4). Teal boxes correspond to RBD^+^ B-cell fractions, which include four samples from CD19^+^CD27^−^RBD^+^ and four from CD19^+^CD27^+^RBD^+^ sorted B-cell subsets; red boxes denote samples of CD19^+^CD27^+^RBD^−^ B cells, *—the difference is statistically significant (*p* < 0.05).

**Figure 3 cells-12-00774-f003:**
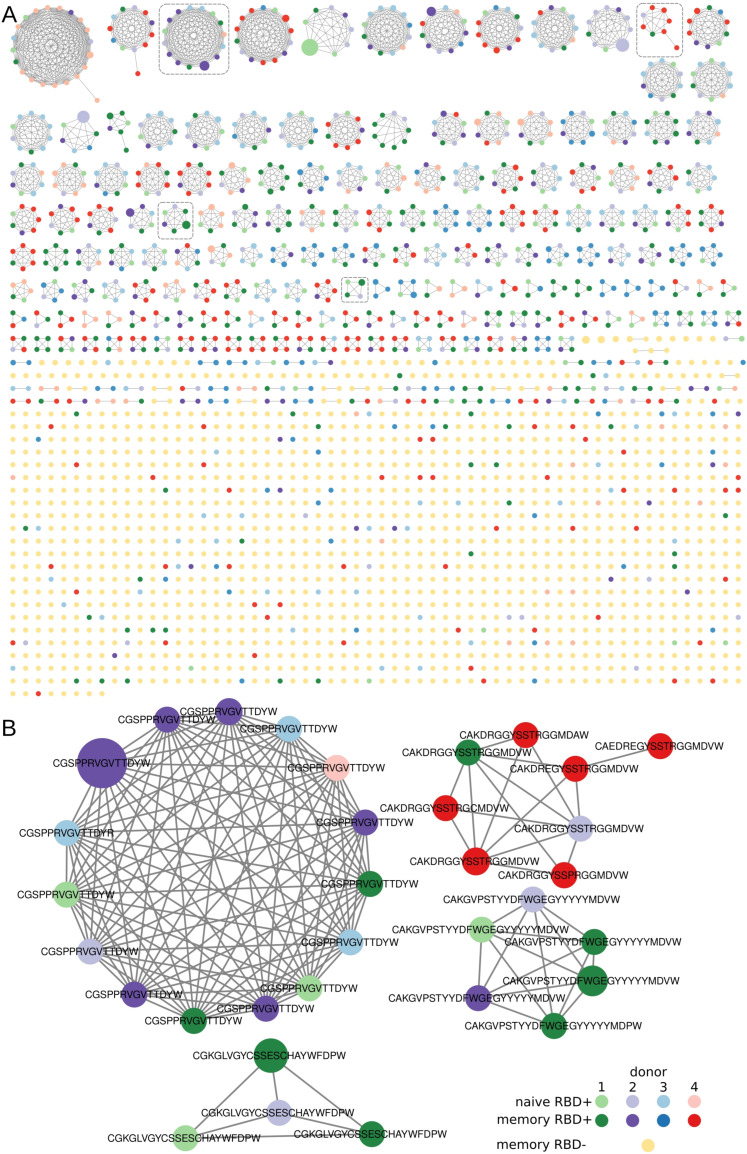
Visualization of clustered sequences. Colored circles represent clonotypes from RBD^+^ (memory, plasmablast, and naïve) and RBD^−^ subsets, respectively (see the legend). The size of the nodes reflects the frequency of the clonotype in our study population. Clonotypes are connected with a line if they share the same *IGHV* segment and have, at most, one amino acid mismatch in the *CDR3* region. (**A**) All clusters and clonotypes. The RBD^+^ sets of sequences have interpatient overlaps, in contrast to sets of sequences belonging to the RBD^−^ B-cell population, which manifested no noticeable clusterization. Gray dashed rectangles highlight the clusters depicted in panel B. (**B**) Clusters that were selected from the donors’ repertoire and contained sequences for subsequent protein synthesis.

**Figure 4 cells-12-00774-f004:**
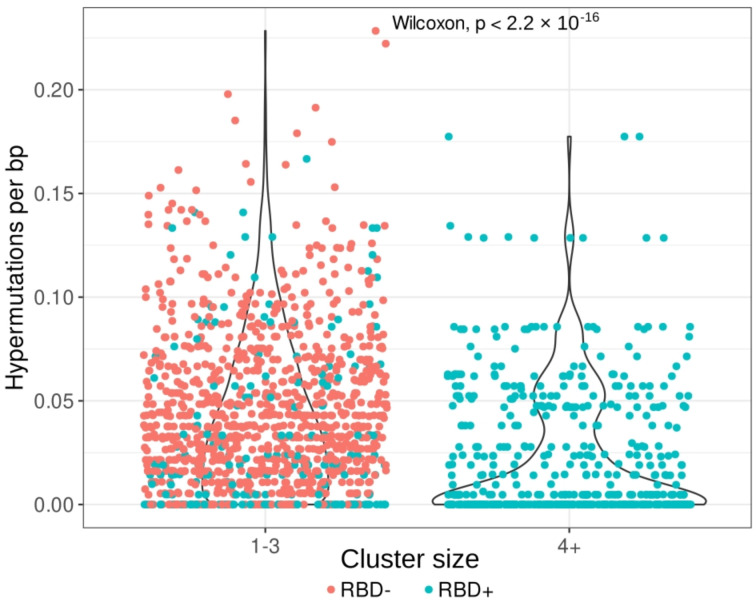
Hypermutation rates of clonotypes from small clusters (1–3 clonotypes, (**left**)) and from large clusters (four and more clonotypes, (**right**)). Clusterization (the same *IGHV* segment and, at most, one amino acid mismatch in the *CDR3* region) was performed on clonotypes pooled from the RBD^−^ subset (red) and from the RBD^+^ subset (teal, both memory and naïve B cells). The average hypermutation rate is significantly lower in clonotypes from large clusters, which consist exclusively of *IGH* clones from RBD^+^ cells.

**Figure 5 cells-12-00774-f005:**
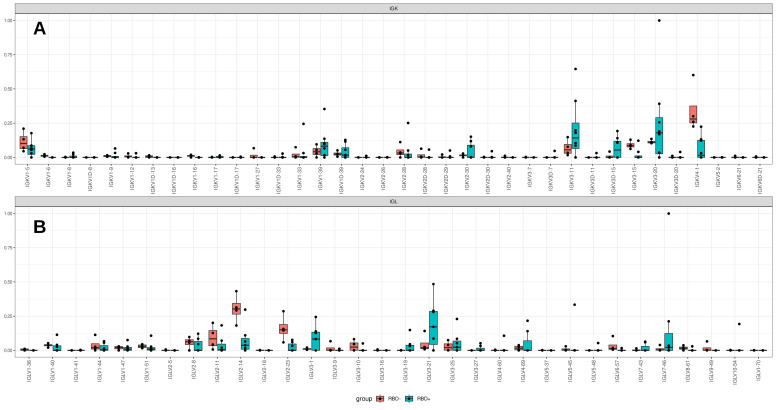
Box plot analysis of pooled Ig sequence data from different patients (*n* = 4). Teal boxes correspond to RBD^+^ cell fractions, which include four samples from CD19^+^CD27^−^RBD^+^ and CD19^+^CD27^+^RBD^+^ sorted B-cell subsets; red box plots represent CD19^+^CD27^+^ RBD^−^ samples. The proportion of light chain ((**A**): kappa, (**B**)): lambda) V usage in the sorted B-cell subtypes is also indicated.

## Data Availability

B cell repertoire sequencing data were deposited in the SRA database under project number PRJNA885120.

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
