# Peer review of "A Rapid Method for Detection of Antigen-Specific B Cells"

_cells, 2023, doi:10.3390/cells12050774_

Round 1

Reviewer 1 Report

P3 L133.-  Check typographical errors throughout the document (Fig A1)

P3 L 139 CD27 is a 50-55 kD type I membrane protein also known as S152 and T14. It is a lymphocyte-specific member of the TNF-receptor superfamily. CD27 is expressed on medullary thymocytes, virtually all mature T cells, some B cells, and NK cells. CD27 binds to CD70 and plays an important role in costimulation of T cell activation, and regulation of B cell differentiation and proliferation. The cytoplasmic domains of CD27 have also been shown to interact with TRAF2 and TRAF5 to elicit NF-κB and SAPK/JNK activation.

CD19 is a 95 kD type I transmembrane glycoprotein also known as B4. It is a member of the immunoglobulin superfamily expressed on B-cells (from pro-B to blastoid B cells, absent on plasma cells) and follicular dendritic cells. CD19 is involved in B cell development, activation, and differentiation. CD19 forms a complex with CD21 (CR2) and CD81 (TAPA-1), and functions as a BCR co-receptor.

P4 L 193 Change the letter of Supplementary Tables and Figures to their corresponding Table S1... and Figure S1...

P10 L380, P11 L381.-  Apart from the limitation of "finding antigen-specific B cells in donors, in the acute infection stage" what other limitations does his method have?

P10 L373.-  Authors should discuss recent work by as:

  ----Wang Z, et al Memory B cell responses to Omicron subvariants after SARS-CoV-2 mRNA breakthrough infection in humans. J Exp Med. 2022 Dec 5;219(12):e20221006. doi: 10.1084/jem.20221006.

 ----Zheng B, et al. B-cell receptor repertoire sequencing: Deeper digging into the mechanisms and clinical aspects of immune-mediated diseases. iScience. 2022 Aug 24;25(10):105002. doi: 10.1016/j.isci.2022.105002.

Author Response

Thank you for attention to our work!

We replaced Fig A1.

We added to Material and Methods section short description of targets for anti-CD19 and CD27 antibodies as following;

P3L132: "Cells were stained with anti CD-19-PE (Biolegend), which is a specific marker for all B lineage cells..."
and
P3L136: "After positive selection the B cells were additionally stained with CD27-BV421 (BioLegend) and RBD-Alexa647 (in house) which is widely used as a marker for memory B cells."

Initially we loaded supplementary figures and tables to Appendix section. We would like to advise the Editor whether we should move it to Supplementary section.

We added the limitations of our approach in Discussion section:

P11L402"Besides the low abundance of specific memory B cells and plasmatic cells in peripheral blood, high similarity and cross-individual sharing of light immunoglobulin chains is an obstacle to functional pairing of immunoglobulin chains. "

We added the suggested references into the article body:

in Introduction:

P2L46:"For the last decade various tools have been developed for Ig repertoire profiling [5].

and Wang et al. in Results and in Discussion:

P11L391:"Additionally, viral subvariants of SARS-CoV-2 elicit immune response mediated by antibodies utilizing different IGHV gene segments [18]."

Reviewer 2 Report

In the manuscript entitled ‘A rapid method for detection of antigen-specific B cells’ by Vakhitova et al., the authors investigated an approach to sort out RBD-specific B cells from COVID-19 patients and collected BCR sequences of these clones for computational analysis. The authors compared the BCR repertoire and VDJ gene usage between RBD+ and RBD- B cells. They also clustered the BCR heavy chain sequences and characterized the IGH clonotypes. Additionally, the authors also compared the usage of V genes usage for light chain analysis. Finally, the authors expressed four selected antibody clones and confirmed the binding to recombinant RBD proteins. This study was generally well-designed and provided additional tools for identifying RBD-specific antibodies from COVID-19 patients. However, several concerns are also raised here.

Major concerns:

1.     For confirming the specificity of the expressed antibody to RBD, data in Table A4 seems insufficient. Please perform a binding ELISA to recombinant RBD with serial dilutions of the four antibodies along with the controls and graph a figure with OD on Y-axis and antibody dilution on X-axis. This will be very helpful to compare antibody binding potency between several clones.   

2.     The discussion section of this manuscript seems to be oversimplified. The authors wrote discussions in many of the results sections. This could be moved to the discussion section. Please make sure to include in the discussion the advantage of the approach in this study and how is it different from previous approaches.  

Minor concerns:

1.     On lines 250 – 258, the authors compared VDJ gene usage with previous studies [25]. This study [25] used samples from 19 patients, but only 4 were used in this manuscript. Please discuss the potential limitation of this low number of samples being studied here.     

Author Response

Thank you for your comments,

As for your remark about ELISA we need extra time to perform it. We have the data for serial dilution but we would like to repeat all dilutions simultaneously.  

we substaintially re-wrote Discussion  section and added 2 references. Please find below:

 “Here we propose an approach that consists of several key steps: RBD-specific sorting of B-cells, BCR repertoire sequencing and subsequent bioinformatics analysis. Based on downstream repertoire analysis we can conclude that RBD-specific BCRs mostly form the largest clusters and can be shared between donors. These features indicate specific clonotypes for selection and subsequent expression in mammalian cells.

In repertoire analysis we observed the heterogeneity of V-gene usage that might be related to individual response to SARS-CoV-2. As demonstrated previously V-gene usage variation among individuals was far larger than differences among disease severity cohorts [25]. Additionally, viral subvariants of SARS-CoV-2 elicit immune response mediated by antibodies utilizing different IGHV gene segments [18].  

Despite the small size of the cohort and the heterogeneity of V usage the proposed clustering analysis of Ig clonotypes from sorted B cells proved to be helpful for finding SARS-CoV-2–specific Ig sequences.

This approach can be applied to other tasks that involve a search for antigen-specific B cells. One of the main problems with finding antigen-specific B cells in peripheral blood is their very low number. We managed to find RBD+ B cells among PBMCs due to the acute stage of infection, when the number of specific cells is the highest. Thus, a limitation of this method for finding antigen-specific B cells is that the donors should be at the acute infection stage. Besides the low abundance of specific memory B cells and plasmatic cells in peripheral blood, high similarity and cross-individual sharing of light immunoglobulin chains is an obstacle to functional pairing of immunoglobulin chains. Nonetheless, the proposed technique may be more applicable to diseases with a sufficiently high local antigen load, for example, a tumor microenvironment (which is often infiltrated by B cells) or sites of chronic inflammation.”

In [25->27] the authors really had a cohort consisted of 19 donors, but only 5 patients had status severe. It can be comparable with our study. Additionally, we discussed it in Discussion section:

P11L387: “In repertoire analysis we observed the heterogeneity of V-gene usage that might be related to individual response to SARS-CoV-2. As demonstrated previously V-gene usage variation among individuals was far larger than differences among disease severity cohorts [27]”.